# Sex Difference in Effectiveness of Early Rhythm- over Rate-Control in Patients with Atrial Fibrillation

**DOI:** 10.3390/jcm11174991

**Published:** 2022-08-25

**Authors:** Dong-Seon Kang, Daehoon Kim, Eunsun Jang, Hee Tae Yu, Tae-Hoon Kim, Hui-Nam Pak, Jung-Hoon Sung, Moon-Hyoung Lee, Pil-Sung Yang, Boyoung Joung

**Affiliations:** 1Division of Cardiology, Department of Internal Medicine, Severance Cardiovascular Hospital, Yonsei University College of Medicine, Seoul 03722, Korea; 2Department of Cardiology, CHA Bundang Medical Center, CHA University, Seongnam 13497, Korea

**Keywords:** atrial fibrillation, early rhythm control, cardiovascular outcome

## Abstract

**Background:** This study aimed to investigate the associations between sex and the relative effect of rhythm control over rate control in patients with atrial fibrillation. **Methods:** We used the National Health Insurance Service database to select patients treated for atrial fibrillation within one year after diagnosis. The primary composite outcome comprised cardiovascular death, ischemic stroke, heart failure hospitalization, or acute myocardial infarction. **Results:** During the mean follow-up (4.9 ± 3.2 years), the benefit of rhythm control over rate control on the primary composite outcome became statistically insignificant after 3 months from atrial fibrillation diagnosis in women while remained steadily until 12 months in men. The risk of primary composite outcome for rhythm control was lower than that for rate control in both sexes if it was initiated within 6 months (men: HR = 0.86, 95%CI = 0.79–0.94; women: HR = 0.85, 95%CI = 0.78–0.93; *P* for interaction = 0.844). However, there was significant interaction between sex and the relative effect of rhythm control if it was initiated after 6 months (men: HR = 0.72, 95%CI = 0.52–0.99; women: HR = 1.32, 95%CI = 0.92–1.88; *P* for interaction = 0.018). **Conclusion:** Rhythm control resulted in lower risk of primary composite outcome than rate control in both sexes; however, the treatment initiation at an earlier stage might be considered in women.

## 1. Introduction

Atrial fibrillation (AF) is associated with increased risks of stroke, congestive heart failure (HF), and mortality [1]. Rhythm control and rate control are representative treatment strategies for atrial fibrillation and previous randomized trials have attempted to demonstrate differences in long-term outcomes between the two strategies. The landmark Atrial Fibrillation Follow-up Investigation of Sinus Rhythm Management (AFFIRM) trial reported no significant differences between these two strategies with respect to mortality and stroke incidence [2,3,4]. Similarly, a meta-analysis of randomized clinical trials comparing rate and rhythm control showed no significant differences in the risk of all-cause death [5]. In contrast, recent studies have demonstrated that early rhythm control (defined as rhythm control initiated ≤12 months from AF diagnosis) compared to rate control in patients with AF is associated with a lower risk of the first primary outcome, comprising stroke, HF hospitalization, acute coronary syndrome, and cardiovascular death [6,7,8].

Many studies highlighten sex differences in the epidemiology, pathophysiology, and prognosis of AF [1]. In this regard, several studies have demonstrated that despite the tendency of women to be more symptomatic compared to men, they are less likely to undergo rhythm control [9,10,11,12]. In women with AF, the use of antiarrhythmic drugs (AADs) were associated with higher rate of life-threatening adverse events [13]. Moreover, female sex was associated with higher AF recurrence rates after radiofrequency ablation compared to male sex, which may influence the effectiveness of AF treatment [14]. However, the effect of sex differences on outcomes of rhythm and rate control has not been well elucidated yet. Similarly, it is not clear whether the effect of timing of treatment initiation (duration from AF diagnosis to the first initiation of rhythm or rate control) on outcomes is affected by sex differences. Therefore, this study was designed to analyze the effect of sex on the comparative effectiveness of early rhythm control over rate control and clarify whether sex makes a difference in the timing of treatment initiation to improve cardiovascular outcomes.

## 2. Methods

### 2.1. Study Design and Population

This retrospective cohort study was based on the National Health Claims Database established by the National Health Insurance Service (NHIS) of Korea, which incorporates the data of 558,147 participants recruited from a total of 5.5 million individuals aged ≥60 years included in the database. 

Appendix A presents the details of this study design. Adults (age ≥18 years) who were treated for AF within one year after AF diagnosis between 1 January 2005 and 31 December 2015, were screened. Inclusion criteria were as follows: individuals aged ≥ 75 years; individuals with a previous transient ischemic attack or stroke; and those who at least met two of the following criteria: age ≥ 65 years, women, hypertension, diabetes mellitus, HF, previous myocardial infarction (MI), or chronic kidney disease [6,8]. Accordingly, patients were excluded from the study if within a six-month period from the initiation of AF treatment, did not receive adequate oral anticoagulants (for at least three months) or died. (Figure 1A).

The Tenth Revision of International Classification of Disease (ICD-10) code I48 was used to define AF. The positive predictive value for AF diagnosis was 94.1% in the NHS database [15]. We adopted a new-user and intention-to-treat design to compare outcomes of rhythm- or rate control. Patients who have never been prescribed the drugs of interest or undergone radiofrequency ablation for AF were regarded as new users. Intention-to-treat with rhythm control was defined as performance of radiofrequency ablation or over three-months’ administration of any AADs within the six-month period since the first prescription. Intention to treat with rate control was defined as a prescription any rate control drugs for at least three months within a six-month period since the first prescription, without pre-scription of AADs and radiofrequency ablation. Accordingly, patients who had received both rhythm- and rate control simultaneously were regarded as the rhythm control group. Claim codes for antiarrhythmic- and rate control drugs, and radiofrequency ablation are demonstrated in Appendix A. To assess the effect of the timing of treatment initiation, patients were divided into two groups as following: AF treatment initiation <6 months group and ≥6 months group after AF diagnosis. 

### 2.2. Outcome and Follow-Up

The primary composite outcome constituted of is chemic stroke, HF hospitalization, acute MI, and cardiovascular death. We also examined the risks of each component of the primary composite outcome. The definition of the outcomes is detailed in Appendix A. The composite safety outcome consisted of all-cause death, intracranial or gastrointestinal bleeding that required hospital admission, or prespecified serious adverse events related to rhythm control. Accordingly, cardiac tamponade, syncope, sick sinus syndrome, atrioventricular block, pacemaker implantation, and sudden cardiac arrest were defined as prespecified serious adverse events related to rhythm control. The study outcomes were followed up from 180 days after the first recorded prescription or procedure until December 31, 2016, or death. Details of the variables are also presented in Appendix A.

### 2.3. Statistical Analysis

Descriptive data were reported as means (standard deviations) for continuous variables and numbers (percentages) for categorical variables. After dividing into two groups according to treatment initiation, overlap weighting based on a propensity score (ps) was used to assess the differences in baseline characteristics between the rhythm- and rate control groups among men and women, respectively. The propensity score, which indicates the probability of being assigned to a rhythm control group, was calculated by logistic regression analysis based on socio-demographic factors, AF duration, year in which treatment was initiated, level of care at which the AF treatment was provided, clinical risk scores, medical history, and concurrent medication use (variables in Table 1). Continuous variables were modelled as cubic spline functions. Appendix A depicted the distribution of propensity scores before and after overlap weighting, respectively. The overlap weight was calculated as ‘1-ps’ in rhythm control groups and as ‘ps’ in rate control groups [16]. A standardized mean difference < 0.1 was considered to indicate acceptable differences in all baseline variables between the two groups. Competing risk regression by the Fine and Gray method was used to consider all-cause death as a competing event when estimating the risks of clinical outcomes [17]. Cofactors with a standardized mean difference of 0.1 or more after weighting were included as covariates in the competing risk regression analysis. Schoenfeld residuals were used to evaluate the proportional hazards assumption and violation of the assumption was not found. To explore the treatment timing-dependent effect of rhythm control on the out-comes, Cox proportional hazards models were fit to the entire weighted study population using an interaction term for the treatment timing after AF diagnosis (modelled as a natural spline) and treatment (rhythm- or rate control). Standard errors were estimated using 1,000 bootstrap replicates. Statistical analyses were performed by SAS, version 9.3 (SAS Institute, Cary, NC, USA) and R version 4.1.0 (The R Foundation, www.R-project.org (accessed on 1 September 2021)).

### 2.4. Sensitivity Analyses

First, one-to-one ps matching (without replacement with a caliper of 0.01) was used instead of overlap weighting. Second, we performed an analysis after including patients treated with AADs as the initial choice of rhythm control. Third, we performed falsification analysis to measure systematic bias in this study by employing 24 pre-specified falsification endpoints, with true hazard ratios of 1.

## 3. Results

### 3.1. Baseline Characteristics

Among 28,049 patients who underwent AF treatment within 1 year from AF diagnosis, 14,383 (51.3%) were men. Compared to men, women were older (68.5 ± 11.4 vs. 66.0±11.2 years, *p* < 0.001) and had a higher CHA2DS2-VASc score (4.3 ± 1.7 vs. 3.4 ± 1.4, *p* < 0.001) (Table 1). Further, the time period between the treatment initiation and AF diagnosis was shorter for women (0.9 ± 2.2 vs. 1.1 ± 2.3 months, *p* < 0.001), and they were less treated with rhythm control (47.6% vs. 49.8%, *p* < 0.001).

Among the initial rhythm control strategies, amiodarone accounted for the largest portion (2874 [44.1%] of 6504 women and 3193 [44.6%] of 7164 men), followed by propafenone and flecainide (Figure 1B). Radiofrequency ablation was performed in 88 (1.4%) of women and 120 (1.7%) of men at the time of enrollment and was eventually per-formed in 294 (4.5%) of women and 530 (7.4%) of men until the end of follow-up, respectively. 

Baseline characteristics of men and women treated with rhythm- or rate control be-fore and after overlap weighting are presented in Table 1 and Table 2. Compared to rate-control patients, rhythm-control patients were younger and tended to have a higher prevalence of comorbidities for both men and women. After weighting, all baseline characteristics were well-balanced between rhythm- and rate control group in both sexes.

### 3.2. Sex Difference of the Primary Composite Outcome according to the Timing of Rhythm Control

The mean follow-up times were 4.9±3.2 years. Cox proportional hazard models with an interaction term showed that women had a linear relationship, wherein the relative effect of rhythm control over rate control on the primary composite outcome became attenuated as the timing of treatment initiation was delayed (Figure 2A, B). Rhythm control was associated with a significantly lower risk of the primary composite outcome compared to rate control if it was initiated within 3 months from AF diagnosis; however, the benefit be-came statistically insignificant after 3 months. On the other hand, in men, relative effect of rhythm control over rate control on the primary composite outcome was maintained until 12 months after AF diagnosis.

In the group with AF treatment initiated within 6 months after the first diagnosis of AF, the risk of primary composite outcome for rhythm control tended to be lower than that of rate control in both the sexes (men: HR = 0.86, 95% CI, 0.79–0.94, P = 0.001; women: HR = 0.85, 95% CI, 0.78–0.93, P < 0.001; P for interaction = 0.844) (Table 3). In the group with AF treatment initiated after 6 months, significant interaction was demonstrated between sex and the relative effect of rhythm control over rate control (men: HR = 0.72, 95% CI, 0.52–0.99, P = 0.045; women: HR = 1.32, 95% CI, 0.92–1.88, P = 0.134; P for interaction = 0.018).

The relative effects of rhythm control over rate control on the individual outcomes are presented in Table 4. Among the individual cardiovascular outcomes, there was a significant interaction between the relative effect of rhythm control over rate control on the prevention of ischemic stroke and sex. 

The relative effects of rhythm control over rate control on safety outcomes are presented in Appendix A. There was a trend of the composite safety outcome towards an increased risk in women and reduced risk in men, irrespective of timing of treatment initiation (<6 months: HR = 0.97 in men, HR = 1.10 in women, *p* for interaction = 0.040; ≥6 months: HR = 0.85 in men, HR = 1.27 in women, *p* for interaction = 0.093).

### 3.3. Sensitivity Analyses

Among the patients in whom AF treatment was initiated ≥6 months, significant interaction between sex and the relative effect of rhythm control over rate control on the primary composite outcome was consistently observed in one-to-one ps matching analysis (Appendix A). Enrollment of patients taking AADs as the initial strategy of rhythm control showed consistent results (Appendix A). In the analyses of 24 falsification endpoints, the 95% CIs of the associations of rhythm control with each end-point covered 1 in 24 (100%) endpoints (Appendix A).

## 4. Discussion

### 4.1. Main Findings

The principal findings of this nationwide cohort study that categorized patients according to sex and the timing of treatment initiation were as follows. First, as treatment initiation was delayed, the relative effect of rhythm control over rate control on primary composite outcome was attenuated gradually in women while remained steadily until 12 months in men. Second, among patients who received AF treatment after 6 months from AF diagnosis, there were significant interactions between sex and relative effects of rhythm control over rate control on the primary composite outcome. Third, compared to rate control, rhythm control showed a trend towards an increased risk of the composite safety outcome in women, irrespective of timing of treatment initiation.

### 4.2. Sex Differences in Benefits and Harms of Rhythm Control

AF is a common arrhythmic disease with a higher prevalence in men than in women; however, stroke and mortality risk are significantly higher in women than in men [18,19]. Sex differences in outcomes of rhythm control over rate control were investigated in subgroup analyses of previous trials. The AFFIRM trial showed that mortality rates between rhythm- and rate control did not differ by sex [3]. In comparison, the RACE trial showed that rhythm control was associated with a higher incidence of the primary outcome compared to rate control in women, not in men [13]. Recently, the EAST-AFNET 4 and Kim et al. reported that in comparison with usual care or rate control irrespective of sex, rhythm control initiated within 12 months from AF diagnosis lowered the risk of the first primary outcome (i.e., ischemic stroke, HF hospitalization, acute MI, and cardiovascular death) [6,8]. However, the aforementioned trials did not show the relationship between the outcome of rhythm control and timing of AF treatment initiation in men and women, respectively.

### 4.3. Earlier Rhythm Control Therapy Is Needed in Women

The present study’s findings show that the relative effects of rhythm control over rate control on the primary composite outcome was reversed in women after 6 months from AF diagnosis. Significant interactions in the group that received AF treatment within 6–12 months from AF diagnosis mainly originated from the interaction between sex and relative effect of rhythm control over rate control on ischemic stroke. In a previous randomized controlled trial, which showed that rhythm control offered no advantage or significant disadvantage for ischemic stroke over rate control irrespective of sex, most patients al-ready had AF for >2 years [20]. In the RACE trial, rhythm control led to more thromboembolic complications in women, whereas the opposite trend was observed in men. However, a recent large cohort study reported that rhythm control was associated with a reduced risk of ischemic stroke when it was prescribed within 7 days from AF diagnosis regardless of sex [21]. This finding also supported the results of this study in the group that received AF treatment <6 months from AF diagnosis.

Precise mechanisms of sex differences in outcomes of rhythm over rate control have not been fully elucidated yet. The possible explanation for the waning of relative efficacy of early rhythm-control therapy is that women are older than men at the initial treatment for AF. This finding is consistent with those of previous reports, although women’s symptoms and quality of life were poorer than those of men. Further, they were referred later and were less likely to undergo rhythm control [9,10,11,12]. However, a significant interaction between sex and the primary composite outcome was still noted even after weighing age and comorbidities. Among patients treated with catheter ablation, women had a significantly smaller mean voltage, slow conduction velocity, and greater proportion of complex fractionated signals in the left atrium compared to men [22]. Since atrial remodeling progresses gradually over time, women may have a narrower window to obtain benefits from rhythm control because they already have more advanced atrial remodeling at the time of AF treatment initiation.

### 4.4. Increased Safety Outcome by Rhythm Control in Women

In this study, compared with men, women had a higher risk of the composite safety outcome and adverse event related to rhythm control. Previous studies have reported comparable results for adverse events related to rhythm control. One study demonstrated that AADs tended to increase risks of torsades de pointes and sick sinus syndrome more in women compared to men [23]. Additionally, as use of catheter ablation has been increased during the last few decades, female sex has become a predictor of in-hospital complications for any cardiac arrhythmia [24]. A large retrospective study reported that women tended to have higher risks of access site complications, cardiac tamponade and pericardial effusions, and postoperative bleeding requiring transfusions [25,26,27]. Therefore, even if rhythm control can be initiated at an earlier stage, the benefit of rhythm control in women with AF must be balanced against the risk of adverse event related to rhythm control.

### 4.5. Study Limitations

This study has several limitations. First, a claims-based database was used; hence, it is not possible to evaluate the changes in AF burden before and after AF treatment, the tar-get heart rate for rate control, and the number of patients who had reached the target heart rate. Moreover, AF diagnosis and treatment strategies were defined by ICD-10 or claim codes only; therefore, it was not possible to obtain the data regarding the AF type (paroxysmal vs. non-paroxysmal), and the presence of symptoms (symptomatic vs. asymptomatic); thus, the role of AF type and the symptom status as contributors to long-term out-comes remain unknown. 

Second, the findings from this observational study cannot establish causality due to unmeasured or residual confounding factors. In this study, the vast majority of patients received warfarin. Among patients treated with warfarin, the higher incidence of stroke in women could be related to a lower time in therapeutic range compared to men [28,29]. The frequency of warfarin use and labile international normalized ratio values also can explain the trend towards higher bleeding events in women in the rhythm control group [28]. Therefore, results in population treated with direct anticoagulants are additionally required. Moreover, uncontrolled lifestyle factors (such as obesity, alcohol intake, and exercise habit) might lead to the detrimental long-term outcomes in patients with AF, and it was not possible to determine their effect. 

Third, radiofrequency ablation was performed as an initial rhythm control strategy in only 1.7% of men and 1.4% of women, which were significantly lower compared to the 7% in the EAST-AFNET 4 trial. The cause of this phenomenon was that the national health insurance had reimbursed the cost of treatment only to patients who were diagnosed as drug-refractory AF or could not maintain AADs due to drug-related side effects, tachycardia-bradycardia syndrome, or other conditions [6]. Considering the superiority of radiofrequency ablation over AADs for maintenance of sinus rhythm, the absence of a reasonable portion of patients treated with ablation might have significantly limited the impact of the outcomes of this study. In addition, the reduced benefit of “rhythm control therapy” in women might be attributable to AAD therapy issues rather than rhythm control strategy, as AADs carries higher risk of proarrhythmia and toxicity compared to both ablation and rate control therapy, particularly in women Therefore, further randomized trials are necessary to reflect the long-term efficacy of ablation strategy [30,31].

Fourth, the specific reasons for choosing rhythm control over rate control, and immediate over delayed initiation of treatment are difficult to be evaluated because these decisions vary by physicians. Accordingly, this ambiguity might have caused potential bias. Nevertheless, the results of the falsification analysis showed that systematic bias was less likely to exist, and sufficient overlap of propensity scores were identified between rhythm- and rate control groups, which proves the balance between the two therapies. 

Fifth, since we excluded patients with AF who did not undergo therapy or who had a history of AF treatment, the proportions of treatment strategies in this study may not reflect the preferences in real-world practice. Sixth, this study enrolled only high-risk patients with a mean CHA2DS2-VASc score of 3.3 using inclusion criteria similar to that of EAST-AFNET 4. Thus, further investigation is warranted to elucidate sex differences in effects of rhythm control over rate control in low-risk patients.

Finally, in this study, the mean period between treatment initiation and AF diagnosis was 1.0 ± 2.2 month and only 5% of the patients were treated between 6 and 12 months after AF diagnosis. Therefore, repeated studies will be required to solidify the conclusion that sex differences influence the outcomes if AF treatment is delayed.

## 5. Conclusions

Among patients who underwent rhythm or rate control within one year after AF diagnosis, lower risk tendency of primary composite outcome was shown in rhythm control than rate control in both sexes. However, as treatment initiation was delayed, the benefit of early rhythm control was attenuated gradually in women, while it was maintained in men. Therefore, in women, rhythm control might be taken into consideration at an earlier stage with a careful assessment of the balance between its benefit and risk of adverse event.

## Figures and Tables

**Figure 1 jcm-11-04991-f001:**
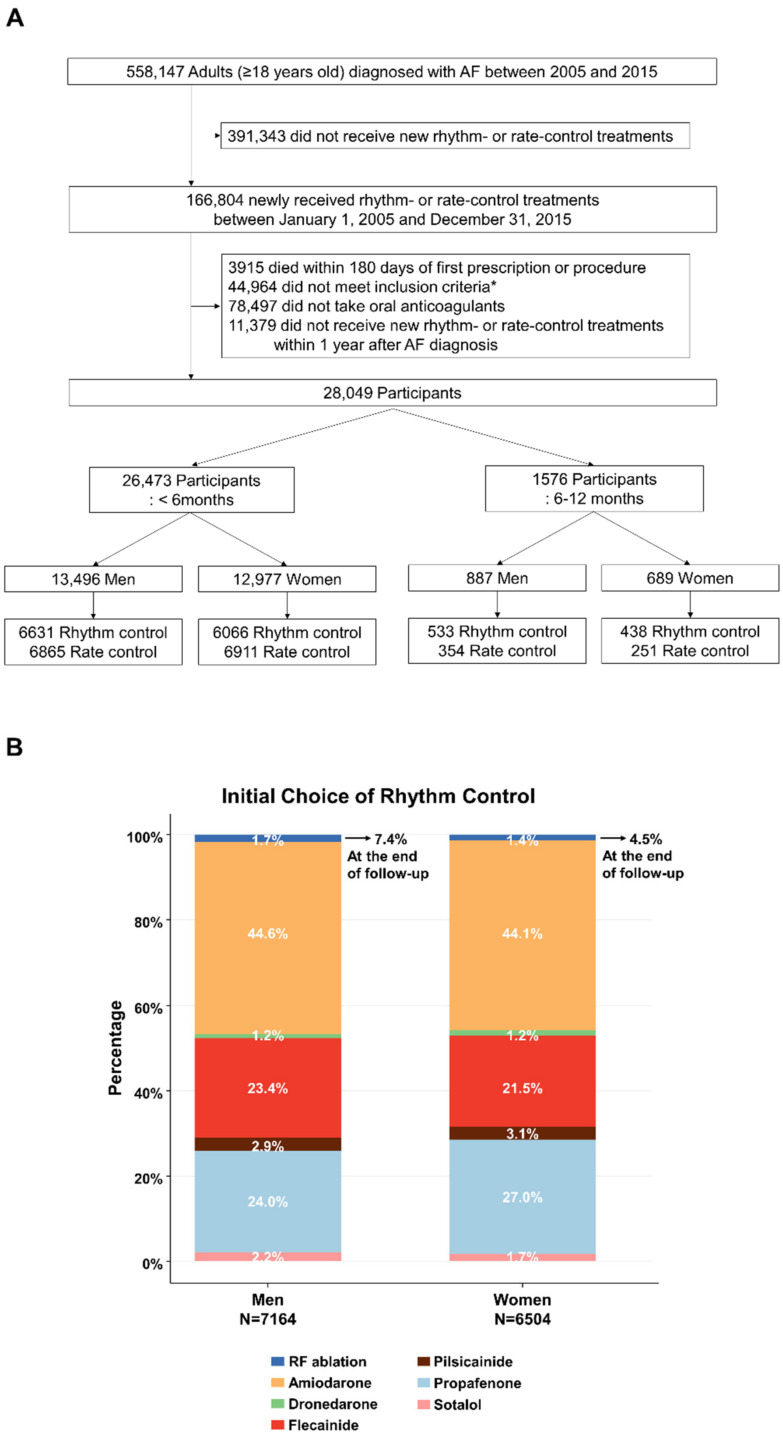
Flow chart. Selection of study participants (**A**) and initial rhythm control strategies according to sex and the timing of treatment initiation (**B**). * Age ≥ 75 years, previous transient ischemic attack or stroke, or two of the following criteria: age ≥ 65 years, women, hypertension, diabetes mellitus, heart failure, previous myocardial infarction, or chronic kidney disease. AF, atrial fibrillation.

**Figure 2 jcm-11-04991-f002:**
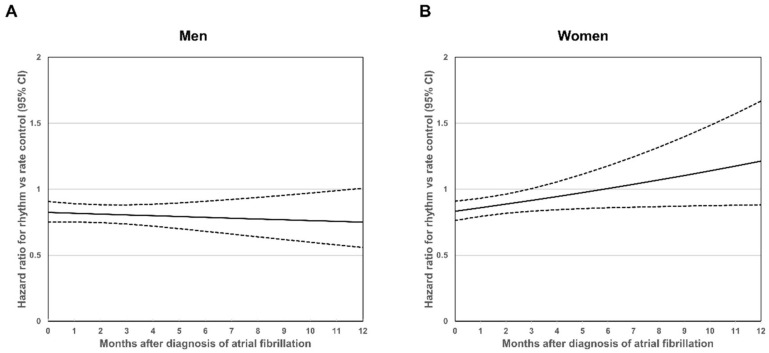
Relationship between treatment timing and primary composite outcome risk. Data shown are within 1 year after the first diagnosis of atrial fibrillation. (**A**) Men. (**B**) Women. Hazard ratio = 1 means an equal risk of outcomes in participants treated with rhythm- and rate-control. Dashed black lines show the 95% confidence interval.

**Table 1 jcm-11-04991-t001:** Baseline characteristics of men and women treated with rhythm- or rate control before overlap weighting.

	Men	Women	*p*-Value	Men				Women			
Treatment Initiation *	<1 Year since AF Diagnosis			<6 Months since AF Diagnosis		6–12 Months since AF Diagnosis		<6 Months since AF Diagnosis		6–12 Months since AF Diagnosis	
	Overall			Rhythm Control	Rate Control	Rhythm Control	Rate Control	Rhythm Control	Rate Control	Rhythm Control	Rate Control
	N = 14383	N = 13666		N = 6631	N = 6865	N = 533	N = 354	N = 6066	N = 6911	N = 438	N = 251
**Sociodemographic**											
Age, years	66.0 (11.2)	68.5 (11.4)	<0.001	65.1 (11.1)	67.0 (11.2)	64.5 (10.5)	67.6 (11.4)	67.6 (11.0)	69.4 (11.6)	66.3 (10.9)	68.9 (11.9)
<65 years	5432 (37.8)	4167 (30.5)	<0.001	2768 (41.7)	2325 (33.9)	228 (42.8)	111 (31.4)	2024 (33.4)	1912 (27.7)	153 (34.9)	78 (31.1)
65–74 year	5532 (38.5)	5064 (37.1)	0.016	2490 (37.6)	2677 (39.0)	223 (41.8)	142 (40.1)	2329 (38.4)	2457 (35.6)	191 (43.6)	87 (34.7)
≥75 years	3419 (23.8)	4435 (32.5)	<0.001	1373 (20.7)	1863 (27.1)	82 (15.4)	101 (28.5)	1713 (28.2)	2542 (36.8)	94 (21.5)	86 (34.3)
AF duration, months	1.1 (2.3)	0.9 (2.2)	<0.001	0.9 (1.4)	0.3 (0.9)	8.8 (1.7)	8.8 (1.8)	0.7 (1.3)	0.3 (0.9)	8.8 (1.8)	8.8 (1.8)
**Enroll year**											
2005–2007	3082 (21.4)	3377 (24.7)	<0.001	1066 (16.1)	1830 (26.7)	95 (17.8)	91 (25.7)	1030 (17.0)	2168 (31.4)	98 (22.4)	81 (32.3)
2008–2010	2814 (19.6)	2648 (19.4)	0.702	1150 (17.3)	1480 (21.6)	104 (19.5)	80 (22.6)	1090 (18.0)	1425 (20.6)	74 (16.9)	59 (23.5)
2011–2013	4163 (28.9)	3790 (27.7)	0.025	2035 (30.7)	1858 (27.1)	166 (31.1)	104 (29.4)	1879 (31.0)	1712 (24.8)	145 (33.1)	54 (21.5)
2014–2015	4324 (30.1)	3851 (28.2)	0.001	2380 (35.9)	1697 (24.7)	168 (31.5)	79 (22.3)	2067 (34.1)	1606 (23.2)	121 (27.6)	57 (22.7)
High tertile of income	6252 (43.5)	5412 (39.6)	<0.001	5513 (83.1)	5098 (74.3)	458 (85.9)	270 (76.3)	5159 (85.0)	5293 (76.6)	379 (86.5)	202 (80.5)
Living in metropolitan areas	6600 (45.9)	6101 (44.6)	0.038	3260 (49.2)	2938 (42.8)	264 (49.5)	138 (39.0)	2936 (48.4)	2858 (41.4)	204 (46.6)	103 (41.0)
**Level of care** **initiating treatment**											
Tertiary	7590 (52.8)	6849 (50.1)	<0.001	4148 (62.6)	2926 (42.6)	354 (66.4)	162 (45.8)	3647 (60.1)	2806 (40.6)	281 (64.2)	115 (45.8)
Secondary	6089 (42.3)	5961 (43.6)	0.031	2276 (34.3)	3494 (50.9)	159 (29.8)	160 (45.2)	2244 (37.0)	3463 (50.1)	144 (32.9)	110 (43.8)
Primary	704 (4.9)	856 (6.3)	<0.001	207 (3.1)	445 (6.5)	20 (3.8)	32 (9.0)	175 (2.9)	642 (9.3)	13 (3.0)	26 (10.4)
**Risk scores**											
CHA2DS2-VASc score	3.4 (1.4)	4.3 (1.7)	<0.001	3.4 (1.4)	3.3 (1.3)	3.6 (1.5)	3.8 (1.4)	4.4 (1.8)	4.2 (1.6)	4.7 (1.7)	4.7 (1.7)
HAS-BLED score †	2.4 (1.1)	2.3 (1.1)	<0.001	2.5 (1.1)	2.3 (1.0)	2.7 (1.1)	2.7 (1.1)	2.4 (1.1)	2.1 (1.1)	2.6 (1.1)	2.6 (1.1)
Charlson comorbidity index	3.5 (2.8)	3.3 (2.8)	<0.001	4.0 (2.8)	2.9 (2.6)	4.7 (2.8)	4.4 (2.9)	4.0 (2.8)	2.6 (2.5)	4.5 (2.7)	4.3 (2.9)
Hospital Frailty Risk score	3.5 (4.8)	3.8 (5.3)	<0.001	3.4 (4.6)	3.5 (4.8)	3.7 (5.1)	5.5 (6.7)	4.0 (5.4)	3.4 (5.1)	4.5 (5.5)	5.7 (7.6)
**Medical history**											
Heart failure	7013 (48.8)	7258 (53.1)	<0.001	3083 (46.5)	3482 (50.7)	290 (54.4)	158 (44.6)	3049 (50.3)	3820 (55.3)	262 (59.8)	127 (50.6)
Heart failure hospitalization	1974 (13.7)	2186 (16.0)	<0.001	778 (11.7)	1100 (16.0)	65 (12.2)	31 (8.8)	852 (14.0)	1223 (17.7)	80 (18.3)	31 (12.4)
Hypertension	10748 (74.7)	10037 (73.4)	0.015	5574 (84.1)	4403 (64.1)	484 (90.8)	287 (81.1)	5107 (84.2)	4317 (62.5)	404 (92.2)	209 (83.3)
Diabetes	4324 (30.1)	3130 (22.9)	<0.001	2214 (33.4)	1840 (26.8)	181 (34.0)	89 (25.1)	1618 (26.7)	1343 (19.4)	111 (25.3)	58 (23.1)
Dyslipidemia	10376 (72.1)	9626 (70.4)	0.002	5340 (80.5)	4312 (62.8)	460 (86.3)	264 (74.6)	4875 (80.4)	4184 (60.5)	379 (86.5)	188 (74.9)
Ischemic stroke	5104 (35.5)	3822 (28.0)	<0.001	2156 (32.5)	2568 (37.4)	183 (34.3)	197 (55.6)	1652 (27.2)	1906 (27.6)	148 (33.8)	116 (46.2)
Transient ischemic attack	1307 (9.1)	1070 (7.8)	<0.001	699 (10.5)	508 (7.4)	70 (13.1)	30 (8.5)	587 (9.7)	396 (5.7)	58 (13.2)	29 (11.6)
Hemorrhagic stroke	301 (2.1)	256 (1.9)	0.203	146 (2.2)	127 (1.8)	15 (2.8)	13 (3.7)	120 (2.0)	120 (1.7)	6 (1.4)	10 (4.0)
Myocardial infarction	1454 (10.1)	1003 (7.3)	<0.001	757 (11.4)	603 (8.8)	64 (12.0)	30 (8.5)	520 (8.6)	413 (6.0)	54 (12.3)	16 (6.4)
Peripheral arterial disease	1641 (11.4)	1442 (10.6)	0.023	937 (14.1)	567 (8.3)	82 (15.4)	55 (15.5)	838 (13.8)	514 (7.4)	68 (15.5)	22 (8.8)
Valvular heart disease	1388 (9.7)	2843 (20.8)	<0.001	673 (10.1)	625 (9.1)	49 (9.2)	41 (11.6)	1082 (17.8)	1612 (23.3)	78 (17.8)	71 (28.3)
Chronic kidney disease	802 (5.6)	525 (3.8)	<0.001	448 (6.8)	286 (4.2)	46 (8.6)	22 (6.2)	320 (5.3)	169 (2.4)	24 (5.5)	12 (4.8)
Hyperthyroidism	1205 (8.4)	1796 (13.1)	<0.001	684 (10.3)	423 (6.2)	76 (14.3)	22 (6.2)	959 (15.8)	722 (10.4)	86 (19.6)	29 (11.6)
Hypothyroidism	1005 (7.0)	1801 (13.2)	<0.001	553 (8.3)	368 (5.4)	66 (12.4)	18 (5.1)	1034 (17.0)	653 (9.4)	90 (20.5)	24 (9.6)
Malignancy	3032 (21.1)	2051 (15.0)	<0.001	1496 (22.6)	1297 (18.9)	142 (26.6)	97 (27.4)	1072 (17.7)	858 (12.4)	78 (17.8)	43 (17.1)
Hypertrophic cardiomyopathy	260 (1.8)	256 (1.9)	0.716	146 (2.2)	95 (1.4)	14 (2.6)	5 (1.4)	160 (2.6)	76 (1.1)	19 (4.3)	1 (0.4)
Sleep apnea	86 (0.6)	17 (0.1)	<0.001	58 (0.9)	24 (0.3)	3 (0.6)	1 (0.3)	10 (0.2)	7 (0.1)	438 (100.0)	251 (100.0)
**Concurrent medication ‡**											
Oral anticoagulant	14383 (100.0)	13666 (100.0)	-	6631 (100.0)	6865 (100.0)	533 (100.0)	354 (100.0)	6066 (100.0)	6911 (100.0)	438 (100.0)	251 (100.0)
Warfarin	12778 (88.8)	12163 (89.0)	0.682	5724 (86.3)	6265 (91.3)	467 (87.6)	322 (91.0)	5175 (85.3)	6365 (92.1)	391 (89.3)	232 (92.4)
Direct oral anticoagulant	1734 (12.1)	1586 (11.6)	0.251	977 (14.7)	651 (9.5)	72 (13.5)	34 (9.6)	935 (15.4)	581 (8.4)	49 (11.2)	21 (8.4)
Beta–blocker	8271 (57.5)	7320 (53.6)	<0.001	3093 (46.6)	4695 (68.4)	237 (44.5)	246 (69.5)	2674 (44.1)	4278 (61.9)	206 (47.0)	162 (64.5)
Non–dihydropyridine CCB	2149 (14.9)	2079 (15.2)	0.536	944 (14.2)	1065 (15.5)	88 (16.5)	52 (14.7)	779 (12.8)	1206 (17.5)	62 (14.2)	32 (12.7)
Digoxin	3659 (25.4)	4342 (31.8)	<0.001	631 (9.5)	2863 (41.7)	59 (11.1)	106 (29.9)	667 (11.0)	3536 (51.2)	53 (12.1)	86 (34.3)
Aspirin	3482 (24.2)	2701 (19.8)	<0.001	1627 (24.5)	1640 (23.9)	127 (23.8)	88 (24.9)	1245 (20.5)	1316 (19.0)	96 (21.9)	44 (17.5)
P2Y12 inhibitor	1372 (9.5)	827 (6.1)	<0.001	672 (10.1)	616 (9.0)	46 (8.6)	38 (10.7)	389 (6.4)	395 (5.7)	30 (6.8)	13 (5.2)
Statin	5524 (38.4)	5002 (36.6)	0.002	2667 (40.2)	2511 (36.6)	211 (39.6)	135 (38.1)	2418 (39.9)	2293 (33.2)	200 (45.7)	91 (36.3)
Dihydropyridine CCB	2459 (17.1)	2147 (15.7)	0.002	1389 (20.9)	879 (12.8)	128 (24.0)	63 (17.8)	1251 (20.6)	781 (11.3)	73 (16.7)	42 (16.7)
ACEi/ARB	8352 (58.1)	7514 (55.0)	<0.001	3746 (56.5)	4102 (59.8)	307 (57.6)	197 (55.6)	3317 (54.7)	3826 (55.4)	244 (55.7)	127 (50.6)
Loop/thiazide diuretics	6646 (46.2)	8029 (58.8)	<0.001	2596 (39.1)	3678 (53.6)	209 (39.2)	163 (46.0)	3039 (50.1)	4605 (66.6)	237 (54.1)	148 (59.0)
K+ sparing diuretics	2844 (19.8)	3399 (24.9)	<0.001	1001 (15.1)	1726 (25.1)	67 (12.6)	50 (14.1)	1128 (18.6)	2121 (30.7)	100 (22.8)	50 (19.9)

Data are presented as means (standard deviations) or n (%). * Duration from AF diagnosis to the first initiation of rhythm- or rate control. ^†^ Modified HAS-BLED = hypertension, 1 point; age > 65 years, 1 point; previous stroke, 1 point; his-tory of bleeding or predisposition, 1 point; liable international normalized ratio, not assessed; alcohol or drug abuse, 1 point; and drug predisposing to bleeding, 1 point. ‡ Defined as a prescription supply of over three months within the six months after the first prescription for antiarrhythmic or rate control drugs or the performance of a radiofrequency ablation for AF. ACEi, angiotensin-converting enzyme inhibitor; AF, atrial fibrillation; ARB, angiotensin II receptor blocker; CCB, calcium channel blocker.

**Table 2 jcm-11-04991-t002:** Baseline characteristics of men and women treated with rhythm- or rate control after overlap weighting.

	Men						Women					
Treatment Initiation *	<6 Months since AF Diagnosis			6–12 Months since AF Diagnosis			<6 Months since AF Diagnosis			6–12 Months since AF Diagnosis		
	Rhythm Control	Rate Control	SMD	Rhythm Control	Rate Control	SMD	Rhythm Control	Rate Control	SMD	Rhythm Control	Rate Control	SMD
	N = 2123	N = 2123		N = 132	N = 132		N = 1912	N = 1912		N = 100	N = 100	
**Sociodemographic**												
Age, years	66.0 (11.1)	66.0 (11.5)	<0.001	66.0 (11.2)	66.0 (12.0)	<0.001	68.7 (11.1)	68.7 (11.8)	<0.001	67.5 (10.1)	67.5 (12.2)	<0.001
<65	823.3 (38.8)	802.2 (37.8)	0.02	49.2 (37.1)	46.6 (35.2)	0.04	569.3 (29.8)	569.5 (29.8)	<0.001	30.2 (30.3)	33.1 (33.2)	0.062
65–74	798.1 (37.6)	809.1 (38.1)	0.011	56.4 (42.6)	55.8 (42.1)	0.009	712.9 (37.3)	699.6 (36.6)	0.014	44.1 (44.2)	36.2 (36.3)	0.161
≥75	501.7 (23.6)	511.8 (24.1)	0.011	26.9 (20.3)	30.0 (22.7)	0.058	629.9 (32.9)	643.1 (33.6)	0.015	25.4 (25.5)	30.4 (30.5)	0.112
AF duration, months	0.6 (1.1)	0.6 (1.2)	<0.001	8.8 (1.8)	8.8 (1.8)	<0.001	0.5 (1.1)	0.5 (1.2)	<0.001	8.8 (1.7)	8.8 (1.8)	<0.001
**Enroll year**												
2005–2007	412.0 (19.4)	412.0 (19.4)	<0.001	28.5 (21.5)	28.5 (21.5)	<0.001	417.0 (21.8)	417.0 (21.8)	<0.001	25.8 (25.9)	25.8 (25.9)	<0.001
2008–2010	404.1 (19.0)	404.1 (19.0)	<0.001	29.2 (22.0)	29.2 (22.0)	<0.001	372.2 (19.5)	372.2 (19.5)	<0.001	21.5 (21.5)	21.5 (21.5)	<0.001
2011–2013	627.5 (29.6)	627.5 (29.6)	<0.001	39.3 (29.7)	39.3 (29.7)	<0.001	553.4 (28.9)	553.4 (28.9)	<0.001	27.4 (27.5)	27.4 (27.5)	<0.001
2014–2015	679.5 (32.0)	679.5 (32.0)	<0.001	35.5 (26.8)	35.5 (26.8)	<0.001	569.5 (29.8)	569.5 (29.8)	<0.001	25.0 (25.1)	25.0 (25.1)	<0.001
High tertile of income	915.0 (43.1)	915.0 (43.1)	<0.001	62.6 (47.2)	62.6 (47.2)	<0.001	785.1 (41.1)	785.1 (41.1)	<0.001	37.6 (37.7)	37.6 (37.7)	<0.001
Living in metropolitan areas	980.4 (46.2)	980.4 (46.2)	<0.001	61.6 (46.5)	61.6 (46.5)	<0.001	869.0 (45.4)	869.0 (45.4)	<0.001	46.1 (46.2)	46.1 (46.2)	<0.001
**Level of care** **initiating treatment**												
Tertiary	1102.7 (51.9)	1102.7 (51.9)	<0.001	74.8 (56.5)	74.8 (56.5)	<0.001	966.1 (50.5)	966.1 (50.5)	<0.001	55.2 (55.4)	55.2 (55.4)	<0.001
Secondary	924.0 (43.5)	924.0 (43.5)	<0.001	49.1 (37.1)	49.1 (37.1)	<0.001	856.4 (44.8)	856.4 (44.8)	<0.001	38.2 (38.3)	38.2 (38.3)	<0.001
Primary	96.4 (4.5)	96.4 (4.5)	<0.001	8.5 (6.4)	8.5 (6.4)	<0.001	89.6 (4.7)	89.6 (4.7)	<0.001	6.3 (6.4)	6.3 (6.4)	<0.001
Risk score												
CHA_2_DS_2_-VASc score	3.4 (1.4)	3.4 (1.4)	<0.001	3.7 (1.5)	3.7 (1.4)	<0.001	4.4 (1.8)	4.4 (1.7)	<0.001	4.7 (1.8)	4.7 (1.8)	<0.001
HAS-BLED score †	2.5 (1.1)	2.5 (1.1)	<0.001	2.7 (1.1)	2.7 (1.1)	<0.001	2.3 (1.2)	2.3 (1.1)	<0.001	2.6 (1.1)	2.6 (1.1)	<0.001
Charlson comorbidity index	3.6 (2.6)	3.6 (2.9)	<0.001	4.5 (2.8)	4.5 (2.9)	<0.001	3.5 (2.6)	3.5 (2.8)	<0.001	4.3 (2.5)	4.3 (3.0)	<0.001
Hospital Frailty Risk Score	3.6 (4.8)	3.6 (4.9)	<0.001	4.4 (5.9)	4.4 (5.8)	<0.001	4.0 (5.4)	4.0 (5.4)	<0.001	5.0 (6.1)	5.0 (6.3)	<0.001
**Medical history**												
Heart failure	1030.7 (48.5)	1030.7 (48.5)	<0.001	68.6 (51.8)	68.6 (51.8)	<0.001	1003.6 (52.5)	1003.6 (52.5)	<0.001	55.2 (55.4)	55.2 (55.4)	<0.001
Heart failure hospitalization	294.9 (13.9)	294.9 (13.9)	<0.001	16.0 (12.1)	16.0 (12.1)	<0.001	310.9 (16.3)	310.9 (16.3)	<0.001	14.0 (14.0)	14.0 (14.0)	<0.001
Hypertension	1653.9 (77.9)	1653.9 (77.9)	<0.001	116.2 (87.7)	116.2 (87.7)	<0.001	1475.9 (77.2)	1475.9 (77.2)	<0.001	88.1 (88.4)	88.1 (88.4)	<0.001
Diabetes	659.6 (31.1)	659.6 (31.1)	<0.001	40.5 (30.6)	40.5 (30.6)	<0.001	468.1 (24.5)	468.1 (24.5)	<0.001	22.3 (22.4)	22.3 (22.4)	<0.001
Dyslipidemia	1592.2 (75.0)	1592.2 (75.0)	<0.001	106.2 (80.2)	106.2 (80.2)	<0.001	1401.3 (73.3)	1401.3 (73.3)	<0.001	81.0 (81.2)	81.0 (81.2)	<0.001
Ischemic stroke	767.8 (36.2)	767.8 (36.2)	<0.001	56.7 (42.8)	56.7 (42.8)	<0.001	557.3 (29.1)	557.3 (29.1)	<0.001	39.6 (39.7)	39.6 (39.7)	<0.001
Transient ischemic attack	194.9 (9.2)	194.9 (9.2)	<0.001	14.7 (11.1)	14.7 (11.1)	<0.001	154.5 (8.1)	154.5 (8.1)	<0.001	11.1 (11.1)	11.1 (11.1)	<0.001
Hemorrhagic stroke	45.3 (2.1)	45.3 (2.1)	<0.001	3.7 (2.8)	3.7 (2.8)	<0.001	38.4 (2.0)	38.4 (2.0)	<0.001	2.0 (2.0)	2.0 (2.0)	<0.001
Myocardial infarction	221.9 (10.5)	221.9 (10.5)	<0.001	13.6 (10.3)	13.6 (10.3)	<0.001	137.5 (7.2)	137.5 (7.2)	<0.001	7.4 (7.5)	7.4 (7.5)	<0.001
Peripheral arterial disease	244.0 (11.5)	244.0 (11.5)	<0.001	22.6 (17.1)	22.6 (17.1)	<0.001	216.0 (11.3)	216.0 (11.3)	<0.001	12.0 (12.0)	12.0 (12.0)	<0.001
Valvular heart disease	207.6 (9.8)	207.6 (9.8)	<0.001	15.4 (11.6)	15.4 (11.6)	<0.001	373.1 (19.5)	373.1 (19.5)	<0.001	22.0 (22.1)	22.0 (22.1)	<0.001
Chronic kidney disease	123.2 (5.8)	123.2 (5.8)	<0.001	8.9 (6.7)	8.9 (6.7)	<0.001	73.7 (3.9)	73.7 (3.9)	<0.001	4.4 (4.4)	4.4 (4.4)	<0.001
Hyperthyroidism	172.7 (8.1)	172.7 (8.1)	<0.001	11.5 (8.7)	11.5 (8.7)	<0.001	247.4 (12.9)	247.4 (12.9)	<0.001	14.6 (14.6)	14.6 (14.6)	<0.001
Hypothyroidism	146.4 (6.9)	146.4 (6.9)	<0.001	9.6 (7.3)	9.6 (7.3)	<0.001	255.4 (13.4)	255.4 (13.4)	<0.001	13.1 (13.2)	13.1 (13.2)	<0.001
Malignancy	459.7 (21.7)	459.7 (21.7)	<0.001	33.1 (25.0)	33.1 (25.0)	<0.001	301.8 (15.8)	301.8 (15.8)	<0.001	17.7 (17.8)	17.7 (17.8)	<0.001
Hypertrophic cardiomyopathy	39.5 (1.9)	39.5 (1.9)	<0.001	2.0 (1.5)	2.0 (1.5)	<0.001	32.6 (1.7)	32.6 (1.7)	<0.001	0.6 (0.6)	0.6 (0.6)	<0.001
Sleep apnea	12.6 (0.6)	12.6 (0.6)	<0.001	0.2 (0.2)	0.2 (0.2)	<0.001	2.7 (0.1)	2.7 (0.1)	<0.001	99.7 (100.0)	99.7 (100.0)	<0.001
**Concurrent medication ‡**												
Oral anticoagulant	2123.1 (100.0)	2123.1 (100.0)	<0.001	132.4 (100.0)	132.4 (100.0)	<0.001	1912.1 (100.0)	1912.1 (100.0)	<0.001	99.7 (100.0)	99.7 (100.0)	<0.001
Warfarin	1880.1 (88.6)	1880.1 (88.6)	<0.001	117.4 (88.7)	117.4 (88.7)	<0.001	1687.5 (88.3)	1687.5 (88.3)	<0.001	90.1 (90.4)	90.1 (90.4)	<0.001
Direct oral anticoagulant	267.0 (12.6)	267.0 (12.6)	<0.001	15.9 (12.0)	15.9 (12.0)	<0.001	236.1 (12.3)	236.1 (12.3)	<0.001	10.1 (10.1)	10.1 (10.1)	<0.001
Beta-blocker	1416.8 (66.7)	1416.8 (66.7)	<0.001	82.0 (61.9)	82.0 (61.9)	<0.001	1194.6 (62.5)	1194.6 (62.5)	<0.001	63.5 (63.6)	63.5 (63.6)	<0.001
Non–dihydropyridine CCB	370.2 (17.4)	370.2 (17.4)	<0.001	25.7 (19.4)	25.7 (19.4)	<0.001	338.3 (17.7)	338.3 (17.7)	<0.001	16.7 (16.7)	16.7 (16.7)	<0.001
Digoxin	445.0 (21.0)	445.0 (21.0)	<0.001	31.5 (23.8)	31.5 (23.8)	<0.001	480.3 (25.1)	480.3 (25.1)	<0.001	23.0 (23.1)	23.0 (23.1)	<0.001
Aspirin	535.3 (25.2)	535.3 (25.2)	<0.001	33.2 (25.1)	33.2 (25.1)	<0.001	390.9 (20.4)	390.9 (20.4)	<0.001	19.6 (19.6)	19.6 (19.6)	<0.001
P2Y_12_ inhibitor	224.8 (10.6)	224.8 (10.6)	<0.001	13.4 (10.1)	13.4 (10.1)	<0.001	123.9 (6.5)	123.9 (6.5)	<0.001	6.2 (6.2)	6.2 (6.2)	<0.001
Statin	868.7 (40.9)	868.7 (40.9)	<0.001	49.6 (37.4)	49.6 (37.4)	<0.001	741.4 (38.8)	741.4 (38.8)	<0.001	42.6 (42.7)	42.6 (42.7)	<0.001
Dihydropyridine CCB	347.0 (16.3)	347.0 (16.3)	<0.001	25.0 (18.8)	25.0 (18.8)	<0.001	297.4 (15.6)	297.4 (15.6)	<0.001	16.1 (16.1)	16.1 (16.1)	<0.001
ACEI/ARB	1229.2 (57.9)	1229.2 (57.9)	<0.001	75.3 (56.9)	75.3 (56.9)	<0.001	1041.3 (54.5)	1041.3 (54.5)	<0.001	52.3 (52.5)	52.3 (52.5)	<0.001
Loop/thiazide diuretic	979.7 (46.1)	979.7 (46.1)	<0.001	60.4 (45.6)	60.4 (45.6)	<0.001	1096.7 (57.4)	1096.7 (57.4)	<0.001	55.9 (56.1)	55.9 (56.1)	<0.001
K+-sparing diuretic	419.4 (19.8)	419.4 (19.8)	<0.001	19.8 (15.0)	19.8 (15.0)	<0.001	447.4 (23.4)	447.4 (23.4)	<0.001	20.7 (20.8)	20.7 (20.8)	<0.001

Data are presented as means (standard deviations) or n (%). * Duration from AF diagnosis to the first initiation of rhythm- or rate control. ^†^ Modified HAS-BLED=hypertension, 1 point; age > 65 years, 1 point; previous stroke, 1 point; his-tory of bleeding or predisposition, 1 point; liable international normalized ratio, not assessed; alcohol or drug abuse, 1 point; and drug predisposing to bleeding, 1 point. ‡ Defined as a prescription supply of over three months within the six months after the first prescription for antiarrhythmic or rate control drugs or the performance of a radiofrequency abla-tion for AF. ACEi, angiotensin-converting enzyme inhibitor; AF, atrial fibrillation; ARB, angiotensin II receptor blocker; CCB, calcium channel blocker; SMD, standard mean difference.

**Table 3 jcm-11-04991-t003:** Relative effect of rhythm control over rate control on primary composite outcome after overlap weighting.

Primary Composite Outcome	Number of Events	Person-Years	IR	Number of Events	Person-Years	IR	Absolute Rate Difference per 100 Person-Years (95% CI)	Hazard Ratio (95% CI)	*p*-Value	*p* for Interaction
**AF treatment (<6 months since AF diagnosis)**										0.844
**Men**	Rhythm control **(N = 2123)**			Rate control **(N = 2123)**						
	461	7905	5.83	521	7586	6.87	−1.03 (−1.83 to −0.24)	0.86 (0.79–0.94)	0.001	
**Women**	Rhythm control **(N = 1912)**			Rate control **(N = 1912)**						
	516	7200	7.17	590	6956	8.48	−1.31 (−2.24 to −0.39)	0.85 (0.78–0.93)	<0.001	
**AF treatment (6–12 months since AF diagnosis)**										0.018
**Men**	Rhythm control **(N = 132)**			Rate control **(N = 132)**						
	30	527	5.80	40	471	8.55	−2.75 (−6.09 to 0.59)	0.72 (0.52–0.99)	0.043	
**Women**	Rhythm control **(N = 100)**			Rate control **(N = 100)**						
	33	392	8.40	26	404	6.46	1.94 (−1.85 to 5.73)	1.32 (0.92–1.88)	0.134	

AF, atrial fibrillation; CI, confidence interval; HR, hazard ratio; IR, incidence rate.

**Table 4 jcm-11-04991-t004:** Relative effect of rhythm control over rate control on individual components of the primary composite outcome after overlap weighting.

	Men				Women				
	IR	IR	Hazard Ratio (95% CI)	*p*-Value	IR	IR	Hazard Ratio (95% CI)	*p*-Value	*p* for Interaction
**AF treatment (<6 months since AF diagnosis)**									
	Rhythm control **(N = 2123)**	Rate control **(N = 2123)**			Rhythm control **(N = 1912)**	Rate control **(N = 1912)**			
Cardiovascular death	1.63	1.93	0.86 (0.73–1.00)	0.053	2.38	2.29	1.05 (0.91–1.21)	0.517	0.063
Ischemic stroke	2.51	2.94	0.87 (0.77–0.99)	0.035	2.65	3.69	0.72 (0.63–0.82)	<0.001	0.036
Hospitalization for HF	2.25	2.81	0.82 (0.71–0.94)	0.004	3.67	4.08	0.90 (0.81–1.01)	0.086	0.271
Acute myocardial infarction	0.30	0.44	0.70 (0.49–0.99)	0.049	0.20	0.29	0.70 (0.46–1.06)	0.091	0.989
**AF treatment (6–12 months since AF diagnosis)**									
	Rhythm control **(N = 132)**	Rate control **(N = 132)**			Rhythm control **(N = 100)**	Rate control **(N = 100)**			
Cardiovascular death	1.67	2.60	0.68 (0.39–1.18)	0.171	1.81	2.03	0.91 (0.48–1.73)	0.772	0.512
Ischemic stroke	2.44	3.46	0.74 (0.47–1.18)	0.208	3.91	2.48	1.63 (0.97–2.73)	0.063	0.027
Hospitalization for HF	2.51	3.94	0.68 (0.43–1.10)	0.114	3.49	3.23	1.08 (0.64–1.81)	0.770	0.196
Acute myocardial infarction	0.22	0.45	0.54 (0.13–2.13)	0.376	0.54	0.70	0.79 (0.23–2.74)	0.716	0.677

AF, atrial fibrillation; CI, confidence interval; HF, heart failure; IR, incidence rate.

## Data Availability

The data presented in the study are openly available from NHIS.

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
