# Peer review of "Sex Difference in Effectiveness of Early Rhythm- over Rate-Control in Patients with Atrial Fibrillation"

_jcm, 2022, doi:10.3390/jcm11174991_

Round 1
Reviewer 1 Report
This is a retrospective study based on the National Health Claims Database of Korea investigating the impact of sex on the efectiveness of early rhythm control over rate control and if the timing of therapy initiation affect the outcome according to sex category.
The study is very wel designed, method and statistical analysis sufficiently described, the results presented clearly and precise and the discussion isconcise and related to the outcomes
The main limitation of the study is the very limited number of patients who underwent ablation. Although this limitation is reported in the relevant section, the absence of a reasonable percentage of patients treated with ablation may have limited significantly the impact of this study outcomes
Author Response
Point 1 This is a retrospective study based on the National Health Claims Database of Korea investigating the impact of sex on the efectiveness of early rhythm control over rate control and if the timing of therapy initiation affect the outcome according to sex category.
The study is very wel designed, method and statistical analysis sufficiently described, the results presented clearly and precise and the discussion isconcise and related to the outcomes
The main limitation of the study is the very limited number of patients who underwent ablation. Although this limitation is reported in the relevant section, the absence of a reasonable percentage of patients treated with ablation may have limited significantly the impact of this study outcomes.
Response 1: Thank you for your dedicate advice. To reflect your opinion, we added the sentences as follow:
Considering the superiority of radiofrequency ablation over antiarrhythmic drug for maintenance of sinus rhythm, the absence of a reasonable portion of patients treated with ablation may have significantly limited the impact of the outcomes of this study. In addi-tion, the reduced benefit of “rhythm control therapy” in women may be attributable to antiarrhythmic drug therapy issues rather than rhythm control strategy, as antiarrhythmic drug therapy carries higher risk of proarrhythmia and toxicity compared to both ablation and rate control therapy, particularly in women Therefore, further randomized trials are necessary to reflect the long-term efficacy of ablation strategy [28,29] (Lines 337–345)

Reviewer 2 Report
Kang and colleagues present this claims database study from the Korean National Health Insurance, examining initial rhythm vs rate control therapy for AF (within 1 year of diagnosis), separated by sex. The primary composite outcome was improved with rhythm control (vs rate control) for men through 12 months from diagnosis, whereas the improvement with rhythm control for women was seen only within the first 6 months (linearly became statistically insignificant by 3 months). The authors conclude that rhythm control therapy for AF in women should be pursued early.
This database-study is well done, with the relevant statistical analyses and the noted limitations. As the authors point out, one main limitation is the fact that only 1.7% of patients underwent ablation, and it is known that antiarrhythmic drug therapy carries higher risk of proarrhythmia and toxicity compared with both ablation and rate control therapy (particularly in women). So perhaps the limitations should explicitly state the possibility that the apparent reduced benefit of "rhythm control therapy" in women could be more due to antiarrhythmic drug therapy issues rather than the rhythm control strategy.
Other comments:
line 145: age for women should be listed first - based on the current wording.
line 200-201 please note this was non-significant
Author Response
Response to Reviewer 2 Comments
Point 1 Kang and colleagues present this claims database study from the Korean National Health Insurance, examining initial rhythm vs rate control therapy for AF (within 1 year of diagnosis), separated by sex. The primary composite outcome was improved with rhythm control (vs rate control) for men through 12 months from diagnosis, whereas the improvement with rhythm control for women was seen only within the first 6 months (linearly became statistically insignificant by 3 months). The authors conclude that rhythm control therapy for AF in women sould be pursued early.
This database-study is well done, with the relevant statistical analyses and the noted limitations. As the authors point out, one main limitation is the fact that only 1.7% of patients underwent ablation, and it is known that antiarrhythmic drug therapy carries higher risk of proarrhythmia and toxicity compared with both ablation and rate control therapy (particularly in women). So perhaps the limitations should explicitly state the possibility that the apparent reduced benefit of "rhythm control therapy" in women could be more due to antiarrhythmic drug therapy issues rather than the rhythm control strategy.
Response 1: Thank you for your dedicate advice. To reflect your opinion, we added the sentences as follow:
Considering the superiority of radiofrequency ablation over antiarrhythmic drug for maintenance of sinus rhythm, the absence of a reasonable portion of patients treated with ablation may have significantly limited the impact of the outcomes of this study. In addi-tion, the reduced benefit of “rhythm control therapy” in women may be attributable to antiarrhythmic drug therapy issues rather than rhythm control strategy, as antiarrhythmic drug therapy carries higher risk of proarrhythmia and toxicity compared to both ablation and rate control therapy, particularly in women Therefore, further randomized trials are necessary to reflect the long-term efficacy of ablation strategy [28,29] (Lines 337–345)
Point 2 line 145: age for women should be listed first - based on the current wording.
Response 2: Thank you for the correction. Following your comment, we revised the sentence as follow:
Compared to men, women were older (68.5±11.4 versus [vs.] 66.0±11.2 years, P<0.001) and had a higher CHA2DS2-VASc score (4.3±1.7 vs. 3.4±1.4, P<0.001). (Lines 172–173)
Point 3 line 200-201 please note this was non-significant.
Response 3: Following your comment, we removed the following sentence:
Especially, the risk of ischemic stroke for rhythm control initiated between 6 and 12 months after AF diagnosis tended to be lower than that of rate control in men, whereas the opposite trend was observed in women (P for interaction=0.027). (Lines 228–231)
